# Board Structure, CEO Equity-Based Compensation, and Financial Performance: Evidence from MENA Countries

Abdullah A. Aljughaiman [1,]*[ ], Abdulateif A. Almulhim [1][ ] and Abdulaziz S. Al Naim [2]

1 Finance Department, School of Business, King Faisal University, Al-Ahsa 31982, Saudi Arabia; abmulhem@kfu.edu.sa
2 Accounting Department, School of Business, King Faisal University, Al-Ahsa 31982, Saudi Arabia; asaudalnaim@kfu.edu.sa
* Correspondence: abjuqhaiman@kfu.edu.sa

**Abstract:** This paper investigates the association between board of director (BOD) structures and CEO equity-based compensation (long-term incentive) for commercial banks (conventional and Islamic banks) in MENA countries. Specifically, we take board size and board independence to measure the board structure. Furthermore, we investigate the influence of board structure on the association between CEO equity-based compensation and financial performance. Moreover, we compare conventional and Islamic banks in testing these relationships. Using a sample of 65 banks in MENA countries for the period between 2009 and 2020, we show a significant positive association between board size and CEO compensation. However, we find the same association between these variables for IBs, but the effect of board size on CEO compensation is less. We also show that board independence is negatively correlated with CEO compensation. Nevertheless, the relationship between board independence and CEO ownership is positive for IBs. For the moderating test, we find that effective board structure provides more incentives to the CEO, leading them to achieve higher financial performance. The Islamic bank's business model (based on Shari'ah principles) contributes to the different influences of board structure on CEO compensation. Our results provide the insight that a strong and effective board is important for managing the executive's compensation system. The findings of this study have implications for financial firms, policymakers, and regulators. Specifically, the study may help in understanding the benefits of different compensation structures relative to different types of financial firms.

**Keywords:** corporate governance; board of directors; CEO compensation; financial performance; banks; MENA

**JEL Classification:** G21; G32; G34





## 1. Introduction

Over the last several decades, corporate governance has become increasingly important around the world. More and more countries have adopted corporate governance codes and principles for achieving best practices. Maher and Andersson (2000) stated that effective corporate governance improves the efficiency, competitive advantage, and effectiveness of companies. The significance of corporate governance mechanisms lies in the fact that they help to ensure that management acts in the best interests of all stakeholders, and they give investors greater confidence by encouraging both transparency and accountability (Mallin 2007). In addition, it has been shown that effective corporate governance can prevent the occurrence of undesired events that derail the implementation of imperative programmes, and it can inculcate a culture of integrity and mitigate an organisation's risks (John et al. 2008) and enhance financial performance (Al-Matari 2022).

One of the most vital mechanisms that corporate governance manages is a CEO's compensation. According to Conyon and He (2012), the agency theory assumes that CEO

compensation is related to performance in order to resolve the moral hazard problems linked to the asymmetric information between managers and owners. CEOs' compensation has been a growing and important area of research in recent years, especially in emerging markets such as Saudi Arabia, Egypt, and Jordan. In the most modern corporations, especially those in the United States, CEO compensation is a very complex and contentious subject, and it is determined by a board of directors via the compensation committee (Frydman and Jenter 2010). The recent attraction to executive compensation topics is a result of the universal economic recession and the growing interests in corporate governance over the recent decade (Alfawareh et al. 2023; Deysel and Kruger 2015). CEOs' compensation refers to the economic reward given to CEOs, and it is generally measured by basic pay, bonuses, and stocks (Shah et al. 2009).

Solomon (2007) states that the board of directors is similar to a heart that needs to be correctly fitted in order to carry out the critical duties of advising and monitoring top management (Coles et al. 2008). Jamali and Mirshak (2007) stated that the corporate governance mechanism depends on the board of directors, since a board's effectiveness has been the essential focus of recent attention. The main role of the board of directors is to oversee management decisions and control and lead their companies so that they are successful (Mallin 2007). A high-performance board must achieve these objectives: introduce strategic themes to assure the firm's growth, assure accountability for the firm, assist to bring about prosperity, and assure that a highly qualified top management team is managing the firm (Andoh et al. 2023; Epstein and Roy 2006). Nguyen and Vo (2020) report that effective corporate governance can enhance a bank's efficiency. However, prior studies argue for two types of compensation; these comprise non-based equity compensation (Ozdemir and Upneja 2012) and equity-based compensation (Li and Kuo 2017). Nevertheless, the previous literature argues that CEO equity-based compensation provides managers with high-powered incentive (e.g., Conyon and He 2012; Li and Kuo 2017). As a result, this study focuses on such a compensation structure.

However, most of the previous literature has examined the association among the board of directors and CEO compensation for only non-financial firms in developed countries. Banks have been ignored in the investigations of this issue. Due to the important role that financial firms play in the economy and their complex business models, it is important to investigate this relationship using a sample of financial firms. Furthermore, emerging countries have different corporate governance characteristics than developed countries. For example, corporate governance codes in the Middle East and North Africa (MENA hereafter) countries rely on the "comply or explain" principle, and the percentage of independent members on boards in these countries is higher than in developed countries. Moreover, most of the previous literature only examines either total compensation (non-equity and equity compensations) or only non-equity compensation when investigating the relationship between the board of directors and CEO compensation.

This paper aims to explore the impact of board structure on CEO equity compensation as well as the influence of board structure on the association between CEO equity compensation and financial performance. By employing a sample of 65 banks in 11 MENA countries for the period between 2009 and 2020, we find a positive and significate significant relationship between a board's size and CEO equity compensation. However, this association is weaker for Islamic banks. Furthermore, we find a negative and significant association between board independence and CEO equity compensation. In contrast, board independence is positively correlated to CEO structure for Islamic banks. This could be a reason for the IBs business model's differences. Furthermore, we show that an effective board of director could provide appropriate incentive for CEOs, leading to increases in the bank's financial performance for both bank types.

We contribute to the previous literature (e.g., Core et al. 1999; Reddy et al. 2015; Sheikh et al. 2018; Vafeas 1999) by investigating the relationship between board structure and CEO equity-based compensation using a sample of commercial banks in MENA countries. Furthermore, prior studies have not investigated the influence of board structure on the

association between CEO equity incentives and financial performance, which we provide in this study. In addition, contrary to previous studies, we contribute to the previous literature by comparing Islamic and conventional banks in terms of the relationship between board structure and CEO equity compensation. It is apparent that this relationship has not been explored in the financial industry. Furthermore, board structure and CEO equity compensation research in MENA countries is also limited.

The remaining sections of this paper are organised as follows: Section 2 shows the literature review and hypotheses. Section 3 presents the data and methodology. The results and the empirical analysis are shown in Section 4, and the final section presents the conclusion of this paper.

## 2. Literature Review and Hypothesis Development

### 2.1. Corporate Governance

Corporate governance focuses on two significant dimensions. The first dimension concentrates on the stewardship and the accountability of corporate governance, i.e., controlling and monitoring the managers' actions and ensuring that their responsibilities are in the shareholders' interests. The second dimension concentrates on providing managers with appropriate incentive schemes in order to avoid managerial opportunism (Keasey and Wright 1993). Previous studies have argued that providing firm managers with incentive contracts helps align their interests with shareholders' interests (Alfawareh et al. 2023). Incentive contracts can be in the form of share ownership, stock options, or the threat of dismissal (Jensen and Meckling 1976; Fama 1980; Shleifer and Vishny 1997).

### 2.2. Board of Directors' Structure and CEO Compensation

According to the agency theory, CEOs might make decisions that serve their own interests. Ross (1973) stated that agency problems between agent and principles might be raised when the agent acts for their own interests. However, an effective board can mitigate this problem by managing the executive's compensation. Jensen and Meckling (1976) showed that executive compensation packages can mitigate the agency problem and reduce agency costs. CEOs are self-interested and might act resourcefully at the cost of shareholders' interests. Therefore, the board of directors is expected to confine and mitigate executive opportunism and align the CEOs' interests with those of shareholders using effective corporate governance mechanisms and by constructing efficient pay contracts that normally link top management executive compensation with firm performance (Sheikh et al. 2018). Nevertheless, prior studies (e.g., Holmström 1979; Shleifer and Vishny 1997; Matolcsy and Wright 2011) report that CEO behaviour and incentives towards maximising the shareholders' wealth are significantly improved if the compensation includes some long-term equity-based compensation. Specifically, shareholder wealth is increased by achieving high financial performance, which might be a major goal for CEOs if their compensation structure relies on equity-based compensation.

Given the above, researchers have discussed the significance of a board's delegation mechanism and how it influences CEO compensation. For instance, Fama (1980) and Fama and Jensen (1983) argued that board characteristics play an essential role in determining CEO compensation. These studies claimed that outside directors should make compensation decisions, as these directors do not have affiliations with the managers of the firm. That is, such directors are more able to make unbiased decisions regarding CEO quality and their efficient compensation, firing, and hiring. On the other hand, some studies argued that outside directors may be less informed or that their monitoring can be excessive (Adams and Ferreira 2007). Jensen (1993) claimed that, in US firms, CEOs may participate in nominating new directors. Such directors may feel obligated to these CEOs.

Moreover, the influence of board structure on CEO compensation has been empirically examined. For instance, Ozkan (2011) found that larger boards with higher independent proportions pay higher compensation to their CEOs. Alfawareh et al. (2023) discusses that corporate governance mechanisms have influence on CEO pay, which supports the agency

theory arguments. Hallock (1997) found that, when the CEO of firm A is a director on the board of firm B, and the CEO of firm B is a director on the board of firm A (interlocking relations), both CEOs obtain high compensation. Core et al. (1999) examined the level of compensation of large US firms. They found that the level of CEO compensation was higher in the following cases: the CEO participated in nominating new directors; directors had little stake in the firm; the CEO was a board chair; the board's size was large. Along the same lines, Cyert et al. (2002) found that, when CEOs held dual roles in firms, they received higher compensation. Grinstein and Hribar (2004) examined the association between the size of the bonuses received by CEOs and their board power. They found that, when the CEO was also the board chair and was involved in the process of nominating new directors, they received a larger bonus. Cahan et al. (2005) used a sample of 80 public sector firms in New Zealand. They found a positive association between board size and CEO compensation but a negative association between board independence and CEO compensation. In addition, they found that CEO duality positively affects CEO compensation.

Chhaochharia and Grinstein (2009) found that CEOs' pay was reduced by around 17% in firms with a minority of independent directors. Ozkan (2011) investigated the association between CEO pay and performance using a sample of 390 non-financial UK firms for the 1999–2005 period. The researcher found that firms with a large board size and a high proportion of independent directors pay higher compensation levels for CEOs. Similar to Ozkan (2011), Kohli (2018) emphasized that there is a significant positive relationship between board size and CEO compensation. Guthrie et al. (2012) found that board independence has no relationship with the CEO's level of pay. However, the compensation committee independence increases the CEO's pay level, but the increase only occurs when the concentration of institutional ownership is high. Reddy et al. (2015) investigated the relationship between board structure and CEO compensation in New Zealand for the 2005–2010 period. They found that board size was positively related to CEO compensation, showing that larger board size led to higher CEO remuneration. However, independent directors had no significant relationship with CEO compensation.

Utilizing a sample of Australian companies for the 2001–2011 period, Nguyen et al. (2016) found that firms with a large board size pay higher CEO compensation. Benkraiem et al. (2017) investigated the role of gender on boards and board independence in determining CEO compensation. They found that both women sitting on as board members and independent directors positively affect CEO compensation. Al-Najjar (2017) investigated the impact of board characteristics on the CEO compensation of firms listed in the Travel and Leisure sector on the FTSE 350. The researcher found that large boards pay lower CEO compensation. This could be justified, as CEOs may not be able to monitor large boards, leading to lower CEO compensation. Another study by Patnaik and Suar (2020) found that a higher number of independent directors on the board of directors who possess the necessary skills and qualifications can have positive effects with respect to CEO compensation. Nevertheless, independent directors have a positive relationship with respect to CEO compensation. Using a sample of non-financial firms listed on the Karachi Stock Exchange over the 2005–2012 period, Sheikh et al. (2018) found that neither board size nor board independence had a relationship with CEO compensation. Furthermore, Jatana (2023) found that the association between a larger proportion of independent directors and CEO compensations is positive.

Interestingly, after reviewing studies on the relationship between board structure and CEO compensation, we observe that there are conflicting results describing this relationship. Based on the arguments above, we develop the following two hypotheses:

**H1.** *There is a significant association between board size and CEO compensation for banks in MENA countries.*

**H2.** *There is a significant association between board independence and CEO compensation for banks in MENA countries.*

*2.3. Islamic Governance and CEO Compensation*

Unlike conventional banks (CBS), which are based on the profit-maximisation principle (Olson and Zoubi 2008), the business model of IBs relies on Shari'ah principles. Specifically, IBs must comply with Shari'ah law. Aljughaiman and Salama (2019) and Trinh et al. (2020a) argue that IBs must share profits and risks. They are not allowed to provide or receive debts with interest (riba) or engage in excessive risks, and they must prevent uncertainty (gharar) and speculation (Abadi and Silva 2020; Kettell 2011). Within this law, IBs design Shari'ah-compliant financial services and products. The existence of these principles adds to the corporate governance in IBs, as there are more norms and duties that have to be achieved and maintained. Specifically, the characteristics of IBs are therefore different from those of CBs, which might also have different roles relative to corporate governance compared to IBs. Both the International Financial Standards Board (IFSB) and prior studies have argued that IBs are subject to considerable restrictions with respect to their business models (Iqbal 2013; Safiullah and Shamsuddin 2018).

Based on the arguments above, the boards of IBs may encounter additional restrictions relative to the options they have in managing bank activities, thus reducing their ability to achieve high performance. (Aljughaiman and Salama 2019; Aljughaiman et al. 2023) argue that the boards of directors in IBs have additional responsibility in assuring banks' activities to be compliant with Shari'ah law. This responsibility may add further restrictions to the board's ability to manage risks, which in turn might lead to different risk-taking behaviours. The board of directors' decisions regarding compensation might differ from those of CBs. Chhaochharia and Grinstein (2009) argue that the board of directors might reduce the CEO's compensation when the firm encounters additional requirements. Shari'ah principles could be considered as an additional requirement that could influence the compensation policy of IBs. On the other hand, Alnasser and Muhammed (2012) and Trinh et al. (2020b) argue that the existence of IB restrictions may add constraints to managers (e.g., CEOs), which might influence their decisions. However, effective corporate governance could reduce this negative influence on the banks' decisions with respect to CEOs. In detail, good corporate governance enhances the CEO's decision making, as it provides guidance (advisory role) and a monitoring role that can improve the bank's financial performance. This in turn increases the CEO's compensation as they achieve good financial performance for the bank. Based on the arguments above, we suggest the following hypotheses:

**H3.** *There is a significant difference in the influence of board size on CEO compensation among Islamic banks and conventional banks.*

**H4.** *There is a significant difference in the influence of board independence on CEO compensation among Islamic banks and conventional banks.*

*2.4. The CG, CEO Compensation, and Financial Performance*

The association between the board of directors, CEO compensation, and bank performance has drawn significant attention in the field of corporate governance. The board of directors plays a crucial role in determining the compensation of the CEO. The agency theory posits that the board, as representatives of the shareholders, should design compensation packages that align the interests of the CEO with those of the shareholders (Jensen and Meckling 1976). In the context of bank performance, the board has the responsibility to determine the appropriate combination of fixed and variable pay as well as the use of long-term incentives (such as stock options) in order to align the CEO's interests with long-term financial performance and risk management (Fahlenbrach and Stulz 2011). A study by Adams and Mehran (2012) found that bank boards with more independent directors were more likely to use performance-based CEO compensation. Another study by Zoghlami

(2021) investigated the effect of CEO compensation on financial performance on the French stock exchange. The author found that CEO compensation is positively associated with financial performance. In contrast, other studies have shown that excessive CEO compensation can lead to increased risk-taking and reduced bank performance (Cheng et al. 2015; Fahlenbrach and Stulz 2011).

**H5.** *The board of directors has a significant influence on the association between CEO ownership and financial performance.*

### 3. Data and Methodology

#### 3.1. Sample

The initial sample of this study comprised 360 banks that were listed in 22 MENA countries during the 2009–2020 period. Our sample period avoids the potential effect of the recent 2007 financial crisis. The sample has been filtered based on similar criteria employed in the banking literature (see Aljughaiman and Salama 2019; Abdelsalam et al. 2016). These criteria are as follows: (a) banks' full annual reports had to be available; (b) CBs with an Islamic window and investment banks were dropped.[1] We ended up with an unbalanced panel data sample containing 65 listed banks (760 bank year observations) located in 11 MENA countries. We obtained the financial data from Bloomberg and *BankScope* databases. We manually collected the corporate governance-level data from the banks' annual reports, which are available on official websites. Country-level variables were obtained from the World Bank's World Development Indicators database.

Appendix A Table A1 presents the sample distributions by bank type and country with 432 observations for CBs and 328 observations for IBs. Kuwait and Bahrain have the highest number of IBs, while the highest number of CBs is concentrated in Jordan. Panel B in the same table shows the key variables and characteristics classified by country. The findings show that banks in Qatar achieve the highest financial performance compared to other banks in the sample, which achieve 2.29% on average, while the lowest financial performance is achieved by banks operating in Bahrain. Banks in Lebanon pay higher long-term compensation to their CEOs compared to banks in other countries in the sample. On the macroeconomic level, we find that Qatar's economic situation outperforms other countries in the sample, since their GDPG is 8.7 on average compared to the lowest (1.4) achieved by performance, which is exhibited by Kuwait.

#### 3.2. Measures of Variables

Following Matolcsy and Wright (2011), we measure CEO compensation by taking the percentage of stock ownership held by a CEO. According to Kim and Lu (2011), stock ownership is a reliable proxy to measure managerial compensation. The corporate governance factor was captured through the board of directors' structure. Specifically, we take two proxies for board effectiveness, which are board size, measured by the number of directors on the board, and board independence, measured by the percentage of independent members on the board (Almulhim 2023).

We also control for a number of firm-specific and country-specific variables. At the firm characteristic level, we control for CEO tenure, which is measured by the number of years the CEO has served in this position. Hou et al. (2013) argue that long-tenured CEOs are very likely to take low equity ownership because they become less engaged in extensive information processing. We also control for institutional ownership, as a higher proportion of institutional ownership might lead to low CEO ownership. Khan et al. (2005) found that a larger percentage of owner concentration is related to a lower level of compensation. Firm size is expected to influence CEO ownership, as a larger firm provides a higher percentage of ownership to the CEO. Thus, we control for the firm size.

A firm's financial performance might affect managerial ownership, since firms tend to provide ownership to executives as an incentive to increase the returns. However, most firms set up an incentive plan for managers when they achieve bad returns. Thus, the

CEO ownership could be affected (Fahlenbrach and Stulz 2011). We measure the firm's financial performance by taking the return on average assets. Furthermore, we control for investment opportunities, measured by Tobin's Q, and leverage, measured by equity to total assets. As our sample includes conventional and Islamic banks, we control for IBs using a dummy variable that takes the value of 1 if the bank is Islamic and zero otherwise. We control for country-specific variables by considering GDP growth. Also, we control for the years fixed effect.

### 3.3. Estimation Methods

Pooled Ordinary Least Squares (OLS) with robust standard errors is used to control for heteroscedasticity. To test for the sensitivities of the results, we use different classifications of control variables. Besides, we employ both the GMM system and a lag model to control for any potential endogeneity issues. We test our hypotheses H1 and H2 by running the following empirical model, as shown in Equation (1):

$$CEOOWNER_{i,j,t} = a_0 + b_1\ BODS_{i,j,t} + b_2\ BODI_{i,j,t} + \gamma * X_{i,j,t} + \delta * GDPGrowth_{j,t} + \varepsilon_{i,j,t} \tag{1}$$

where *CEOOWNER* is the CEO stock ownership of bank, *BODS* is the board size, *BODI* is the percentage of independent members, *X* is the matrix of the bank-level control variables, *GDPGrowth* is the matrix of country-level macroeconomic variables, and $\varepsilon$ is the error term.

Regarding the hypotheses H3 and H4, we run the following empirical model, as shown in Equation (2):

$$CEOOWNER_{i,j,t} = a_0 + b_1\ BODS_{i,j,t} + b_2\ BODI_{i,j,t} + b_3\ (BODS * IB)_{i,j,t} + b_4\ (BODI * IB)_{i,j,t} + \gamma * X_{i,j,t} + \delta \\ * GDPGrowth_{j,t} + \varepsilon_{i,j,t} \tag{2}$$

where $(BODS * IB)$ is the interaction term between board size and Islamic banks, and $(BODI * IB)$ is the interaction term between the percentage of independent members and Islamic banks. The rest of the variables are described in Equation (1).

For hypothesis H5, we run the following empirical model, as shown in Equation (3):

$$Performance_{i,j,t} = a_0 + b_1\ (BOD * CEOOWNER)_{i,j,t} + b_2\ (BOD * CEOOWNER * IB)_{i,j,t} + b_3\ BOD_{i,j,t} + \\ b_4\ CEOOWNER_{i,j,t} + \gamma * X_{i,j,t} + \delta * GDPGrowth_{j,t} + \varepsilon_{i,j,t} \tag{3}$$

where *Performance* is the bank return on average asset, $(BOD * CEOOWNER)$ is the interaction term between board of directors index and CEO ownership, and $(BOD * CEOOWNER * IB)$ is the interaction term between board of directors index, CEO ownership, and Islamic banks. The rest of the variables are described in Equation (1).

## 4. Results and Discussion

### 4.1. Descriptive Statistics

We present our descriptive statistics in Table 1. Table 1 shows the mean and the distributional characteristics of all the variables used in our regression. The mean value of CEO stock ownership is 0.42%. Moving to the financial performance of banks in our sample, the mean value of ROAA is 1.19%, where the max return that banks achieve is 4.46%. The mean values of board size (BODS) and board independence (BODI) are 9.9 and 0.36, respectively. This means that the average board size of banks in our sample is 10 members on the board, and 36% of them are independent members. Interestingly, banks in our sample do not appoint new CEOs until they have served for approximately 6 years in this position. For bank characteristics, we find that the average size of the banks in our sample is 15.63, whereas the smallest bank size has a value of 11.17. The mean value of bank growth opportunity is 1.43%, where the average value of the equity to total assets (ETA) is equal to 14.28%. Importantly, 43.3% of our sample is classified as Islamic banks.

**Table 1.** Descriptive statistics.

| Variable | N | mean | p50 | sd | min | max | CBs | IBs | *t*-test |
|---|---|---|---|---|---|---|---|---|---|
| **CEOOWNER** | 736 | 0.421 | 0.0007 | 1.905 | 0.000 | 7.690 | 0.646 | 0.143 | 4.677 *** |
| **BODS** | 734 | 9.976 | 10.00 | 1.874 | 5.000 | 16.00 | 10.292 | 9.368 | 4.304 *** |
| **BODI** | 760 | 0.365 | 0.4285 | 0.245 | 0.000 | 1.000 | 0.346 | 0.390 | −2.539 ** |
| **CEOT** | 760 | 6.450 | 4.000 | 8.038 | 0.000 | 54.00 | 6.988 | 5.74 | 2.97 ** |
| **INSTITO** | 760 | 0.723 | 0.480 | 0.236 | 0.000 | 0.996 | 0.549 | 0.951 | −1.805 * |
| **SIZE** | 760 | 15.636 | 15.617 | 1.738 | 11.17 | 24.149 | 15.7 | 15.552 | 1.842 * |
| **ETA** | 760 | 14.289 | 12.432 | 7.251 | 6.430 | 43.210 | 13.034 | 15.941 | −7.596 *** |
| **ROAA** | 760 | 1.197 | 1.429 | 1.239 | −2.581 | 4.460 | 1.377 | 0.96 | 6.319 *** |
| **GROWO** | 760 | 1.436 | 1.068 | 26.13 | 0.0002 | 600.79 | 1.044 | 1.056 | −1.824 * |
| **IB** | 760 | 0.431 | 0.000 | 0.492 | 0.000 | 1.000 | - | - | - |
| **GDPG** | 760 | 4.028 | 3.978 | 4.784 | −7.076 | 26.17 | - | - | - |

**Notes:** The table presents descriptive statistics of all variables used in the regression models. It also presents the *t*-test for the mean value for both samples (CBs and IBs banks). * $p < 0.10$; ** $p < 0.05$; *** $p < 0.01$ (two-tailed test). CEOOWNER: CEO stock ownership, BODS: board size, BODI: board independence, CEOT: CEO tenure, INSTITO: institutional ownership, SIZE: bank size, ETA: equity to total assets, ROAA: return on average assets, GROWO: growth opportunity, IB: Islamic banks, GDPG: GDP growth.

The *t*-test in Table 1 presents a comparison between IBs and CBs across all main variables. The results show that the mean value of CEO ownership in CBs is significantly higher compared to CEO ownership in IBs. This indicates that Islamic banks provide less compensation and long-term incentives (CEO ownership) to their CEOs. Interestingly, the mean value of the board size of CBs is higher than the average board size of IBs, while IBs appoint a higher number of independent members on their board of directors compared to CBs. Furthermore, CEOs in CBs serve longer in their position compared to IBs, which is 7 years compared to 5 years, respectively. Institutional shareholders own more shares in Islamic banks than the institutional shareholders of conventional banks. In contrast, IBs maintain higher equity as reserves, and CBs achieve higher performance.

Table 2 presents the correlation matrix using the Pearson pairwise correlation for all the variables. This allows us to check for any significant intervariable correlations. The results of this table show that there is no high degree of cross-correlation between the key variables. This confirms that there is no problem of multicollinearity among the regressors. Furthermore, the correlation between board size (BODS) and the CEO stock ownership (CEOOWNER) is positively significant, whereas the relationship between BODI and CEO compensation is negatively significant.

**Table 2.** Correlation matrix.

| Variable | CEOOWNER | BODS | BODI | CEOT | INSTITO | SIZE | ROAA | ETA | Growo | IB | GDPG |
|---|---|---|---|---|---|---|---|---|---|---|---|
| **CEOOWNER** | 1.0 | | | | | | | | | | |
| **BODS** | 0.16 * | 1.0 | | | | | | | | | |
| **BODI** | −0.08 * | −0.25 * | 1.0 | | | | | | | | |
| **CEOT** | −0.07 | 0.13 * | 0.00 | 1.00 | | | | | | | |
| **INSTITO** | −0.08 | 0.13 * | −0.22 * | 0.09 * | 1.00 | | | | | | |
| **SIZE** | 0.17 * | −0.06 | 0.14 * | 0.02 | −0.16 * | 1.00 | | | | | |
| **ROAA** | 0.02 | 0.08 | −0.13 * | 0.03 | 0.04 | 0.28 * | 1.00 | | | | |
| **ETA** | −0.09 * | −0.18 * | 0.17 * | −0.11 * | −0.13 * | −0.30 * | −0.19 * | 1.00 | | | |
| **GROWO** | −0.06 | −0.12 * | 0.04 | −0.03 | −0.09 * | 0.30 * | 0.14 * | −0.11 * | 1.00 | | |
| **IB** | −0.13 * | −0.14 * | 0.12 * | −0.02 | −0.10 * | −0.13 * | −0.19 * | 0.25 * | 0.04 | 1.00 | |
| **GDPG** | −0.01 | −0.07 | −0.15 * | −0.07 | −0.10 * | −0.02 | 0.13 * | 0.10 * | 0.08 * | 0.01 | 1.00 |

**Notes:** The table shows the Pearson pairwise correlation matrix for all variables used in the analysis. * $p < 0.10$ (two-tailed test). CEOOWNER: CEO stock ownership, BODS: board size, BODI: board independence, CEOT: CEO tenure, INSTITO: institutional ownership, SIZE: bank size, ETA: equity to total assets, ROAA: return on average assets, GROWO: growth opportunity, IB: Islamic banks, GDPG: GDP growth.

### 4.2. Empirical Results

4.2.1. BOD Characteristics and CEO Ownership

Table 3 provides the results for CEOOWNER, where we regress CEO stock ownership on the board structure variables. For sensitivity purposes, column 1 only shows the results that were obtained after regressing the main variables. Columns 2 and 3 show the results after controlling for firm and government variables and years fixed effects, respectively.

**Table 3.** Results of regression between CG and CEO compensation.

| | (1) | (2) | (3) |
| --- | --- | --- | --- |
| **Variables** | **CEOOWNER** | **CEOOWNER** | **CEOOWNER** |
| **BODS** | 0.156 *** | 0.196 *** | 0.203 *** |
| | (0.040) | (0.054) | (0.082) |
| **BODI** | −0.875 *** | −0.957 *** | −1.037 *** |
| | (0.306) | (0.748) | (0.760) |
| **BODS*IB** | −0.093 *** | −0.175 *** | −0.175 *** |
| | (0.017) | (0.140) | (0.141) |
| **BODI*IB** | 1.178 *** | 1.188 *** | 1.217 *** |
| | (0.336) | (0.916) | (0.921) |
| **CEOT** | | −0.010 | −0.010 |
| | | (0.010) | (0.010) |
| **InstitO** | | −0.001 | −0.002 |
| | | (0.542) | (0.549) |
| **Size** | | 0.432 *** | 0.432 *** |
| | | (0.079) | (0.143) |
| **ETA** | | −0.032 *** | −0.032 *** |
| | | (0.011) | (0.012) |
| **ROAA** | | −0.189 *** | −0.181 *** |
| | | (0.121) | (0.124) |
| **GROWO** | | −0.577 *** | −0.577 *** |
| | | (0.254) | (0.266) |
| **IB** | | −0.625 | −0.520 |
| | | (1.571) | (1.579) |
| **GDPG** | | 0.037 ** | 0.037 ** |
| | | (0.030) | (0.030) |
| **Constant** | −0.596 | −6.232 *** | −6.200 *** |
| | (0.385) | (2.403) | (2.407) |
| **Year Effects** | **NO** | **NO** | **YES** |
| **Observations** | **712** | **712** | **712** |
| **R-squared** | **0.068** | **0.188** | **0.198** |

**Note:** The table presents regression results for board structure and CEO compensation for the period 2009–2020. Heteroscedasticity-robust standard errors are in parentheses. ** $p < 0.05$; *** $p < 0.01$. CEOOWNER: CEO stock ownership, BODS: board size, BODI: board independence, CEOT: CEO tenure, INSTITO: institutional ownership, SIZE: bank size, ETA: equity to total assets, ROAA: return on average assets, GROWO: growth opportunity, IB: Islamic banks, GDPG: GDP growth.

The BODS has a significant positive association with CEOOWNER at the 1% level across all columns. This suggests that the larger the board of directors is, the higher the percentage of CEO stock ownership is. This finding is in line with prior studies in the literature (see, Cahan et al. 2005; Reddy et al. 2015; Nguyen et al. 2016). These studies argue that the board of directors is expected to restrain and soften executive opportunism and associate the CEOs' interests with shareholders' interests by constructing an effective pay contracts policy that links top executive compensation with firm financial performance (Sheikh et al. 2018). According to agency theory, larger boards can be less effective in disciplining and monitoring CEOs, leading to less oversight and potentially higher compensation demands from CEOs (Fama and Jensen 1983).

In addition, the BODI is negatively and significantly associated with the CEOOWNER at the 1% level across all columns. This shows that lower proportions of independent members on the board are related to higher CEO ownership. Our result is consistent

with Cahan et al. (2005) who found a negative association between board independence and CEO compensation. According to Fama (1980) and Fama and Jensen (1983), outside directors are more able to make unbiased decisions regarding CEO quality and their efficient compensation, firing, and hiring. Moreover, independent members, with their lack of personal ties to the corporation, are traditionally considered more objective in monitoring CEO compensation and performance. Having fewer independent directors might weaken this monitoring mechanism, potentially creating room for CEOs to negotiate higher pay packages (Fama and Jensen 1983).

For Islamic banks results, our second independent variable (the interaction between board size and IBs) has a negative and significant relationship with CEOOWNER at the 1% level. However, since our main board size variable is strongly positive at a 0.15 coefficient, and the BODS*Islamic is −0.09, this indicates that the board size in IBs increases the CEO ownership as well, although the influence is weaker than in CBs. Regarding the interaction between board independence (BODI) and IB, there is a positive and significant association between the two variables at the 1% level. That is, larger percentages of independent members on the board are related to higher CEO ownership in IBs across all the columns. This indicates that, unlike CBs, independent members in IBs seem to increase the CEO ownership. As we discussed previously, Islamic banks have different business models that could lead to different influences of BOD composition on CEO compensation. Although the CBs model is based on the risk-shifting concept, the Islamic banks model is based on profit and risk sharing (Olson and Zoubi 2008; Aljughaiman and Salama 2019).

In terms of control variables, we find that bank size is significantly and positively associated with CEO compensation. This means that larger banks provide more compensation to their CEOs. In contrast, return on assets, capital ratio, and growth opportunities are negatively associated with CEO compensation. That is, banks with higher returns on assets, capital ratio, and growth opportunities tend to pay less for CEOs. Furthermore, GDP growth (GDPG) has a positive association with CEO compensation, which indicates that banks in countries with higher GDP growth pay more compensation to their CEOs.

### 4.2.2. BOD Characteristics and CEO Ownership (Robustness Check)

Prior studies debate that the research on corporate governance and financial performance may be influenced by endogeneity problems and therefore may employ traditional techniques; for example, OLS may not be sufficient (Wintoki et al. 2012). In this section, Table 4 re-examines the relationship between board structure and CEO compensation after controlling for endogeneity using the lag approach and GMM. Previous studies argue that these methods can solve three types of endogeneity, namely, unobserved heterogeneity, simultaneity, and dynamic endogeneity (Wintoki et al. 2012; Almulhim 2022). Table 4 re-examines the relationship between board structure and CEO compensation after controlling for endogeneity using the lag approach and GMM. Column 1 shows the results using the lag approach, while column 2 presents the findings using the GMM method. AR1, AR2, and Hansan assure that our GMM model is valid. The results are consistent with the main estimation results.

**Table 4.** Robustness check: controlling for endogeneity using lag approaches and GMM.

|  | (1) | (3) |
|---|---|---|
| **Variables** | **Lag** | **GMM** |
| **Dv(t-1)** |  | 0.123 * |
|  |  | (0.072) |
| **BODS** | 0.184 ** | 0.184 ** |
|  | (0.080) | (0.080) |
| **BODI** | −2.809 *** | −1.576 *** |
|  | (0.745) | (0.585) |
| **BODS*IB** | −0.009 * | 0.294 * |
|  | (0.122) | (0.174) |
| **BODI*IB** | 1.755 ** | 0.279 * |
|  | (0.856) | (0.982) |
| **CEOT** | −0.0169 * | 0.019 |
|  | (0.00922) | (0.028) |
| **InstitO** | −1.629 ** | −2.657 ** |
|  | (0.640) | (1.229) |
| **Size** | 0.521 *** | 0.224 |
|  | (0.136) | (0.212) |
| **ETA** | −0.034 ** | 0.026 ** |
|  | (0.014) | (0.011) |
| **ROAA** | −0.185 | 0.105 |
|  | (0.133) | (0.104) |
| **GROWO** | −1.050 *** | 0.697 ** |
|  | (0.268) | (0.272) |
| **IB** | −1.057 | −3.313 ** |
|  | (1.366) | (1.623) |
| **GDPG** | −0.001 | −0.015 ** |
|  | (0.030) | (0.007) |
| **Constant** | −4.989 ** | −3.718 |
|  | (2.271) | (3.896) |
| *Year effects* | Yes | Yes |
| *Observations* | 703 | 679 |
| *R-squared* | 0.294 |  |
| *AR(1) test (p-value)* |  | 0.05 |
| *AR(2) test (p-value)* |  | 0.731 |
| *Hansen test of over-identification (p-value)* |  | 0.881 |

**Note:** The table presents regression results for board structure and CEO compensation for the period 2009–2020 using lag and GMM estimations to control for endogeneity. Heteroscedasticity-robust standard errors are in parentheses. * $p < 0.10$; ** $p < 0.05$; *** $p < 0.01$. CEOOWNER: CEO stock ownership, BODS: board size, BODI: board independence, CEOT: CEO tenure, INSTITO: institutional ownership, SIZE: bank size, ETA: equity to total assets, ROAA: return on average assets, GROWO: growth opportunity, IB: Islamic banks, GDPG: GDP growth.

### 4.2.3. BOD Characteristics and CEO Ownership (Additional Analysis)

In this section, we provide additional analysis by investigating the relationship between BOD structure and CEO compensation for firms in the sample that did not change their CEO (see Table 5). The results are also in line with the main estimation results of Table 3.

**Table 5.** Additional analysis: regression between BOD structure and CEO compensation for sample that did not change CEO.

| Variables | (1) CEOOWNER |
|---|---|
| BODS | 0.154 * |
| | (0.092) |
| BODI | −2.678 *** |
| | (0.831) |
| BODS*IB | −0.042 * |
| | (0.168) |
| BODI*IB | 1.959 ** |
| | (0.945) |
| CEOT | −0.018 * |
| | (0.010) |
| InstitO | −1.481 ** |
| | (0.579) |
| Size | 0.451 *** |
| | (0.161) |
| ETA | −0.039 *** |
| | (0.013) |
| ROAA | −0.182 |
| | (0.139) |
| GROWO | −0.800 *** |
| | (0.281) |
| IB | −0.818 |
| | (1.867) |
| GDPG | 0.023 |
| | (0.034) |
| Constant | −4.144 |
| | (2.657) |
| Year effects | YES |
| Observations | 682 |
| R-squared | 0.250 |

**Note:** The table presents regression results for board structure and CEO compensation for the period 2009–2020 after we sub-sample the firms that did not change CEO over our period of the study. Heteroscedasticity-robust standard errors are in parentheses. * $p < 0.10$; ** $p < 0.05$; *** $p < 0.01$. CEOOWNER: CEO stock ownership, BODS: board size, BODI: board independence, CEOT: CEO tenure, INSTITO: institutional ownership, SIZE: bank size, ETA: equity to total assets, ROAA: return on average assets, GROWO: growth opportunity, IB: Islamic banks, GDPG: GDP growth.

### 4.2.4. BOD Characteristics, CEO Ownership, and Financial Performance

Table 6 presents the results of testing our H5, which investigates the impact of BOD characteristics on the association between CEO ownership and financial performance. We specifically create interaction variables by multiplying board structure and CEO ownership. We utilize principal component analysis using board size and independence to create a board structure index. Principal component analysis allows us to effectively obtain a decomposition value of the correlation matrix of director structures (following Aljughaiman and Salama 2019; Ellul and Yerramilli 2013). This allows us to use the eigenvector in the decomposition as a single main factor in our study. Using principal component analysis provides key benefits for measuring the board of directors mechanisms, which allows us to avoid the subjective elimination of any characteristic or the subjective judgment of the influence of these categories (Tetlock 2007).

We also take the interactions between board structure, CEO ownership, and an Islamic bank dummy variable to capture the influence in the Islamic banks sample. However, the results show that the interaction coefficient variable of BOD structure and CEO ownership has a significant and positive association with a bank's financial performance. This indicates that board structure influences the CEO compensation in a way that makes the financial performance increase. Furthermore, the interaction variable that captures the Islamic

sample shows no significant association, which indicates that the results are not different for the Islamic sample.

**Table 6.** Regression results for BOD, CEO, and performance.

| Variables | (1) ROAA |
|---|---|
| BOD_CEO | 0.031 ** |
| | (0.016) |
| BOD_CEO_IB | 0.053 |
| | (0.044) |
| BOD | −0.114 *** |
| | (0.029) |
| CEOOWNER | −0.049 ** |
| | (0.022) |
| BOD_IB | 0.115 *** |
| | (0.032) |
| CEOT | 0.008 *** |
| | (0.002) |
| InstitO | −0.004 |
| | (0.004) |
| Size | 0.317 *** |
| | (0.037) |
| ETA | −0.017 * |
| | (0.010) |
| GROWO | 0.130 |
| | (0.091) |
| IB | −1.444 *** |
| | (0.333) |
| GDPG | 0.088 *** |
| | (0.010) |
| Constant | −3.928 *** |
| | (0.580) |
| Year effects | YES |
| Observations | 712 |
| R-squared | 0.384 |

**Note:** The table presents regression results for the effect of board structure on the association between CEO compensation and financial performance for the period 2009–2020 using OLS approach. Heteroscedasticity-robust standard errors are in parentheses. * $p < 0.10$; ** $p < 0.05$; *** $p < 0.01$. CEOOWNER: CEO stock ownership, BODS: board size, BODI: board independence, CEOT: CEO tenure, INSTITO: institutional ownership, SIZE: bank size, ETA: equity to total Assets, ROAA: return on average assets, GROWO: growth opportunity, IB: Islamic banks, GDPG: GDP growth.

## 5. Conclusions

Board structure is one of the most important mechanisms for controlling the agency problem in firms. However, many researchers have excluded financial firms from their sample due to their different characteristics (e.g., high leverage and complex business models). More importantly, there is scant research on CEO equity compensation in financial firms. Thus, this paper contributes to the extant literature by examining this issue using a financial firm sample. Specifically, we investigated the influence of board structure on CEO equity compensation. Furthermore, we examined the impact of board structure on the association between CEO equity compensation and financial performance. The study's sample comprised 65 listed banks in MENA countries over a period of 12 years from 2009 to 2020.

The findings of this paper are that board size is positively correlated to CEO equity compensation and that board independence is negatively correlated to CEO compensation. However, we found that board size has a weaker positive influence on CEO compensation for IBs. In addition, the relationship between board independence and CEO compensation is positive. The Shari'ah principles add more restrictions on the board member activity,

leading to different influences on CEO compensation. In addition, we found that an effective board of directors provides more incentive in regard to CEO compensation, wisely leading to increases in financial performance. This is consistent with the agency theory and the idea that board structure could operate as a controlling mechanism to manage executives' compensation.

Overall, this study has implications for financial firms, policymakers, and regulators. The findings shown in this study can provide direction and guidance to regulators who are responsible for managing financial systems in MENA countries and the top management of financial companies. Our findings are relevant in the following manner: these firms need to have an effective board of directors that can mitigate agency costs and enhance corporate performance by implementing an appreciative reward scheme for the CEO.

Our study has some limitations; for example, we only focused on CEO equity compensation. Future studies can therefore employ more aspects of CEO compensation, such as bonuses and salaries, in order to explore its impact on financial performance. Moreover, our sample covered the period from 2009 to 2020, and we observed that a board's size has a positive association with CEO compensation, whereas board independence is negatively correlated with CEO compensation. Future studies may examine this association before and during COVID-19.

**Author Contributions:** Conceptualization, A.A.A. (Abdullah A. Aljughaiman) and A.A.A. (Abdulateif A. Almulhim); methodology, A.A.A. (Abdullah A. Aljughaiman); software, A.A.A. (Abdullah A. Aljughaiman); validation, A.A.A. (Abdullah A. Aljughaiman), A.A.A. (Abdulateif A. Almulhim), and A.S.A.N.; formal analysis, A.A.A. (Abdullah A. Aljughaiman); investigation, A.A.A. (Abdulateif A. Almulhim); resources, A.S.A.N.; data curation, A.A.A. (Abdulateif A. Almulhim); writing—original draft preparation, A.A.A. (Abdullah A. Aljughaiman); writing—review and editing, A.A.A. (Abdulateif A. Almulhim); visualization, A.A.A. (Abdullah A. Aljughaiman) and A.S.A.N.; supervision, A.A.A. (Abdullah A. Aljughaiman); project administration, A.A.A. (Abdulateif A. Almulhim); funding acquisition, A.A.A. (Abdulateif A. Almulhim). All authors have read and agreed to the published version of the manuscript.

**Funding:** This work was supported by the Deanship of Scientific Research, Vice Presidency for Graduate Studies and Scientific Research, King Faisal University, Saudi Arabia [Grant No. GRANT5619].

**Informed Consent Statement:** Not applicable.

**Data Availability Statement:** The data are available upon request.

**Conflicts of Interest:** The authors declare no conflicts of interest.

**Appendix A**

**Table A1.** Sample Distributions.

| Panel A | | | |
|---|---|---|---|
| Observations | | | |
| Country | CBs | IBs | Full Sample |
| BH | 24 | 60 | 84 |
| EG | 36 | 24 | 60 |
| JO | 120 | 24 | 144 |
| KW | 48 | 60 | 108 |
| LB | 48 | 0 | 48 |
| OM | 36 | 16 | 52 |
| PL | 12 | 24 | 36 |
| QA | 60 | 36 | 96 |
| SA | 0 | 48 | 48 |
| TN | 24 | 0 | 24 |
| UAE | 24 | 36 | 60 |
| Total | 432 | 328 | 760 |

**Table A1.** *Cont.*

| Panel B | | | |
| --- | --- | --- | --- |
| Key Variables Classified by Country | | | |
| Country | ROA | CEOW | GDPG |
| BH | 0.360 | 0.285 | 3.602 |
| EG | 1.829 | 0.000 | 3.196 |
| JO | 1.259 | 0.011 | 3.048 |
| KW | 0.634 | 0.000 | 1.473 |
| LB | 1.185 | 4.680 | 3.778 |
| OM | 0.589 | 0.000 | 4.454 |
| PL | 1.136 | 0.779 | 7.525 |
| QA | 2.299 | 0.225 | 8.793 |
| SA | 1.661 | 0.139 | 4.533 |
| TN | 1.223 | 0.000 | 2.213 |
| UAE | 1.039 | 0.064 | 2.851 |
| Total | 1.198 | 0.422 | 4.029 |

**Notes:** The final sample employs unbalanced panel data of 65 listed banks (760 bank year observations) operating in 11 MENA countries. Panel B shows the key variables of the study classified by country.

## Notes

[1]  Conventional banks with Islamic windows are banks that provide products that are compliant with Shari'ah (Beck et al. 2013). This type of bank does not provide a separate financial report for the Islamic products window (Čihák and Hesse 2010), thus we excluded them from our sample.

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
