# Peer review of "Board Structure, CEO Equity-Based Compensation, and Financial Performance: Evidence from MENA Countries"

_ijfs, doi:10.3390/ijfs12010013_

Round 1

Reviewer 1 Report

Comments and Suggestions for Authors

Thank you for submitting this study on the relationship between boards of directors and certain outcomes at both conventional and Islamic banks in 11 MENA countries. This study is a simple analysis of a few different associations. While the study is relatively sound as far as determining these associations, and while I do find the overall context relevant and interesting, I found the presentation lacking and misleading.

First, this study has nothing to do with CEO Compensation - despite the title of the paper, despite the first 4-5 pages of the paper and despite what you claim in building up your first 4 hypotheses. You do not study CEO Compensation at all; your analyses include CEO Ownership, which is a vastly different metric than CEO Compensation. The Kim & Lu (2011) paper you cite to support to use ownership as your measure of compensation is for a very specific context, which is vastly different from yours. There might be a correlation between compensation and ownership, but ownership has never been  proxy for compensation; if you want to study compensation, as you suggest in your first 4 pages, just use a compensation variable (or use 3-5 different compensation variables, as the literature has many available measures to construct).  I do not care whether you study compensation or ownership, as either association could be informative, but you cannot motivate your paper with compensation and then ignore compensation altogether. (The descriptive statistics for CEOOWNER make it clear that this is a problematic variable by itself, even without pretending it's a proxy for compensation.)

The most interesting aspect of this study is the comparison of conventional banks to Islamic banks. Since 43% of your sample are IBs, you have great potential to dig into these comparisons and tell some interesting stories. And, since your full sample is relatively small (65 firms across 11 countries), you could easily perform a clinical analysis comparing conventional banks and IBs. However, you never did this. The only comparisons you make are thru the dummy variables in your regressions. This is, of course, a statistical comparison, but it ignores what may be more nuanced differences and comparisons. Why do you not present descriptive statistics comparing Its to conventional banks? Why isn't financial performance included in the main regressions (including an interactive term with type of bank)? What are the key structural and performance differences between these types of firms? I wanted to see more context and storytelling, not just regressions.

I also would have liked to see some context about the 11 countries, potentially including more information about the GDPG matrix variable. Again, with only 11 countries and 65 banks, you could easily dig into these data to better tell a story about how the 65 banks are distributed across the 11 countries and how macroeconomic factors may influence the bank-level relationships. Ignoring the fact that you have 11 different countries weakens the study.

In your endodgeneity tests, it appears you are just going through some checklist of common treatments for endogeneity, without providing any rationale or support for why you are doing these tests - nor any explanation of the endogeneity problems they are fixing in your study. If your study suffers from significant endogeneity issues, then why do you present any OLS results? That is, if you know your OLS results are so biased and so inconsistent that you have to control for endogeneity, then we should ignore them completely. I would just like to have more information about why you are doing these tests and what problems they are solving.

In your final analysis, you include a governance index determined my some approach to principal component analysis. I don't understand why you chose this approach - nor do I understand what governance variables might be in that governance index. Did you just use size and independences - or did you include other variables? If you just included those two variables, why did you feel the need to create some mysterious black box index instead of analyzing the variables directly? Can you sit down with the board of directors and explain their governance index score, as presented by the PCA, to help them understand the strengths and weaknesses of their governance structure? Creating a mysterious black box variable for the sake of statistical analysis will always weaken the relevance of your study; while, in some cases, it may strengthen the rigor of your study, I do not think it does so in your small sample study with 65 banks and 11 countries. We - as scholars and as boards of directors - need more transparency and clarity, not less. As such, your analysis on the relationship between BOD, CEO and performance tells me little.

I do think you have a nice context for a contribution; but I am not convinced by either the motivation nor the analyses in the current study. I would like to finish reading a paper and know exactly why you performed the study and what I learned from the study, but I cannot say I did in this case. I would again encourage you to begin your motivation by thinking of what you would tell the board of directors about your study if you had 10 minutes to talk to them. They will not care about p-values and endogeneity; they want to know why you performed this study and how they should respond to it. I think that story is what's missing from the current research - and it's where the biggest improvements are possible.

Comments on the Quality of English Language

In general, the quality of English is fine. I circled probably 20-25 edits that need to be made with respect to the writing, but most of these are minor and do not affect the understanding or interpretation of the story.

Author Response

Reviewer 1

Thank you for submitting this study on the relationship between boards of directors and certain outcomes at both conventional and Islamic banks in 11 MENA countries. This study is a simple analysis of a few different associations. While the study is relatively sound as far as determining these associations, and while I do find the overall context relevant and interesting, I found the presentation lacking and misleading.

Response: thanks for your constructive comments. We improved the paper following your guidance. Please find our responses to the comments below.

First, this study has nothing to do with CEO Compensation - despite the title of the paper, despite the first 4-5 pages of the paper and despite what you claim in building up your first 4 hypotheses. You do not study CEO Compensation at all; your analyses include CEO Ownership, which is a vastly different metric than CEO Compensation. The Kim & Lu (2011) paper you cite to support to use ownership as your measure of compensation is for a very specific context, which is vastly different from yours. There might be a correlation between compensation and ownership, but ownership has never been proxy for compensation; if you want to study compensation, as you suggest in your first 4 pages, just use a compensation variable (or use 3-5 different compensation variables, as the literature has many available measures to construct).  I do not care whether you study compensation or ownership, as either association could be informative, but you cannot motivate your paper with compensation and then ignore compensation altogether. (The descriptive statistics for CEOOWNER make it clear that this is a problematic variable by itself, even without pretending it's a proxy for compensation.)

Response: We thank the reviewer for this comment, as we scan the literature, we find that CEO compensation structure is classified to two main categories, these are non-equity compensation and equity-based compensation. Previous literature supports the equity-based compensation since it is more related to long-term incentive and align more with shareholders interests. Therefore, we amend our paper across the whole article accordingly to reflect what we want to examine precisely. Please see section 1, p2-3, highlighted in red.

The most interesting aspect of this study is the comparison of conventional banks to Islamic banks. Since 43% of your sample are IBs, you have great potential to dig into these comparisons and tell some interesting stories. And, since your full sample is relatively small (65 firms across 11 countries), you could easily perform a clinical analysis comparing conventional banks and IBs. However, you never did this. The only comparisons you make are thru the dummy variables in your regressions. This is, of course, a statistical comparison, but it ignores what may be more nuanced differences and comparisons. Why do you not present descriptive statistics comparing Its to conventional banks? Why isn't financial performance included in the main regressions (including an interactive term with type of bank)? What are the key structural and performance differences between these types of firms? I wanted to see more context and storytelling, not just regressions.

Response: Following the reviewer suggestion, we added additional table showing comparisons for the key variables between CBs and IBs using t-test. Please see table 1 and the discussion of this table in section 4.1 highlighted in red. Also see appendix A1- panel A for more statistics .

I also would have liked to see some context about the 11 countries, potentially including more information about the GDPG matrix variable. Again, with only 11 countries and 65 banks, you could easily dig into these data to better tell a story about how the 65 banks are distributed across the 11 countries and how macroeconomic factors may influence the bank-level relationships. Ignoring the fact that you have 11 different countries weakens the study.

Response: Following the reviewer suggestion, we added additional table showing the countries characteristics. Please see appendix A1- panel B and the discussion of this table in section 3.1 highlighted in red.

In your endodgeneity tests, it appears you are just going through some checklist of common treatments for endogeneity, without providing any rationale or support for why you are doing these tests - nor any explanation of the endogeneity problems they are fixing in your study. If your study suffers from significant endogeneity issues, then why do you present any OLS results? That is, if you know your OLS results are so biased and so inconsistent that you have to control for endogeneity, then we should ignore them completely. I would just like to have more information about why you are doing these tests and what problems they are solving

Response: The following statement was added to section 4.2.2. BOD characteristics and CEO ownership (robustness check).

“Prior studies have debated that the research on corporate governance and financial performance may be influenced by endogeneity problems and therefore employing traditional techniques, such as OLS may not be sufficient (Wintoki et al., 2012, Akbar et al., 2017). In this section, Table 4 re-examines the relationship between board structure and CEO compensation after controlling for possible endogeneity using the lag approach and GMM. Previous studies argued that these methods can solve three types of endogeneity, namely unobserved heterogeneity, simultaneity and dynamic endogeneity (Wintoki et al., 2012, Akbar et al., 2017).”

In your final analysis, you include a governance index determined my some approach to principal component analysis. I don't understand why you chose this approach - nor do I understand what governance variables might be in that governance index. Did you just use size and independences - or did you include other variables? If you just included those two variables, why did you feel the need to create some mysterious black box index instead of analyzing the variables directly? Can you sit down with the board of directors and explain their governance index score, as presented by the PCA, to help them understand the strengths and weaknesses of their governance structure? Creating a mysterious black box variable for the sake of statistical analysis will always weaken the relevance of your study; while, in some cases, it may strengthen the rigor of your study, I do not think it does so in your small sample study with 65 banks and 11 countries. We - as scholars and as boards of directors - need more transparency and clarity, not less. As such, your analysis on the relationship between BOD, CEO and performance tells me little.

Response: thanks for this comment, we add additional elaborating on the reason of using such index. Please see section 4.2.4, pp.13 highlighted in red.

I do think you have a nice context for a contribution; but I am not convinced by either the motivation nor the analyses in the current study. I would like to finish reading a paper and know exactly why you performed the study and what I learned from the study, but I cannot say I did in this case. I would again encourage you to begin your motivation by thinking of what you would tell the board of directors about your study if you had 10 minutes to talk to them. They will not care about p-values and endogeneity; they want to know why you performed this study and how they should respond to it. I think that story is what's missing from the current research - and it's where the biggest improvements are possible.

Response: we thank the reviewer for this comment, we amend the introduction to make the motivation and the story of the paper more pronounced.

Comments on the Quality of English Language

In general, the quality of English is fine. I circled probably 20-25 edits that need to be made with respect to the writing, but most of these are minor and do not affect the understanding or interpretation of the story.

Response: We proofread the paper again using the MDPI English services. Please see the certificate attached.

Reviewer 2 Report

Comments and Suggestions for Authors

1- The Introduction section is good, but it does not indicate the samples used, the timeframe of the study, or the main findings of the paper. In research papers, the Introduction should summarise the research, including the primary issue or point of contention and the research aims/objectives, contribution, motivation, methods, and findings.

2- Literature Review is very important in research:  This section is one of the most critical sections of the research that shows the research gap and helps the researcher cover this gap the researcher failed to list the research gap, shows the importance of research and the means to find solutions to the research problem. I recommend that researchers repeat this section in a professional manner. Moreover, the authors did not consider recent studies related to the topic like 2021,2022, 2023. Therefore, i strongly recommend that the authors rearrange and support this section.

3- A discussion of the results is required. Why is an emerging economy and not any other country in the region that has similar characteristics (or is this country an individual compared to others in the region)? In conclusion, the author(s) did not provide consistent arguments to support empirical research on emerging economies in this field, which seriously affects the value of this study.

4- The Conclusion section is clear, but it would be better if the authors further expanded the conclusion by offering answers to the research questions on the paper. They then discuss the implications of the study (methodological, managerial, policymakers, and investors). In addition, they must discuss the research limitations in detail and provide an outline for future research.

5- In the conclusions, i introduce the recent updated bibliography and comment on it in light of the literature. In this regard, please include the following.

(2022). Do corporate governance and top management team diversity have a financial impact among financial sector? A further analysis. Cogent Business & Management, 9(1), 2141093.

(2023). Board Characteristics and performance of listed firms in Ghana. Corporate Governance: The International Journal of Business in Society23(1), 43-71.

(2022). Board of directors’ attributes and corporate outcomes: A systematic literature review and future research agenda. International Review of Financial Analysis84, 102424.

6- Finally, I am not a native speaker, but I suggest author/s to perform professional proofreading.

Comments on the Quality of English Language

Finally, I am not a native speaker, but I suggest author/s to perform professional proofreading.

Author Response

Reviewer 2

1- The Introduction section is good, but it does not indicate the samples used, the timeframe of the study, or the main findings of the paper. In research papers, the Introduction should summarise the research, including the primary issue or point of contention and the research aims/objectives, contribution, motivation, methods, and findings.

Response: we thank the reviewer for this comment, we amend the introduction to make the motivation and the story of the paper more pronounced. Please see section 1 highlighted in red

2- Literature Review is very important in research:  This section is one of the most critical sections of the research that shows the research gap and helps the researcher cover this gap the researcher failed to list the research gap, shows the importance of research and the means to find solutions to the research problem. I recommend that researchers repeat this section in a professional manner. Moreover, the authors did not consider recent studies related to the topic like 2021,2022, 2023. Therefore, i strongly recommend that the authors rearrange and support this section.

Response: we thank the reviewer for this comment, we amend the LR section and include some recent literature in both introduction and LR. Please see section 2 highlighted in red

3- A discussion of the results is required. Why is an emerging economy and not any other country in the region that has similar characteristics (or is this country an individual compared to others in the region)? In conclusion, the author(s) did not provide consistent arguments to support empirical research on emerging economies in this field, which seriously affects the value of this study.

Response: Following the reviewer suggestion, we added additional table showing the countries characteristics. Please see appendix A1- panel B and the discussion of this table in section 3.1 highlighted in red.

4- The Conclusion section is clear, but it would be better if the authors further expanded the conclusion by offering answers to the research questions on the paper. They then discuss the implications of the study (methodological, managerial, policymakers, and investors). In addition, they must discuss the research limitations in detail and provide an outline for future research.

Response: We thank the reviewer for this comment. We amend the conclusion as suggested. please see section 5, heighted in red

5- In the conclusions, i introduce the recent updated bibliography and comment on it in light of the literature. In this regard, please include the following.

(2022). Do corporate governance and top management team diversity have a financial impact among financial sector? A further analysis. Cogent Business & Management, 9(1), 2141093.

(2023). Board Characteristics and performance of listed firms in Ghana. Corporate Governance: The International Journal of Business in Society23(1), 43-71.

(2022). Board of directors’ attributes and corporate outcomes: A systematic literature review and future research agenda. International Review of Financial Analysis84, 102424.

 Response: we include the suggested references in the paper.

6- Finally, I am not a native speaker, but I suggest author/s to perform professional proofreading.

Comments on the Quality of English Language

Finally, I am not a native speaker, but I suggest author/s to perform professional proofreading.

Response: We proofread the paper again using the MDPI English services. Please see the certificate attached.

Reviewer 3 Report

Comments and Suggestions for Authors

The paper examines a well-researched factors in the corporate governance filed. However, I am convinced that the large sample size and the comparison between the commercial banks and Islamic banks constitute an original contribution for this research. Therefore, I would suggest the following comments for your consideration.

When you mention acronyms, be sure that it has been fully defined earlier in the manuscript (e.g., IBs).

Make sure that you follow the journal requirement in presenting the manuscript.

The abstract should capture the research implications, in short words.

Add MENA region to the keywords.

The introduction section is well written, however, you have to use recent literature. The literature you used are so outdated.

In your hypotheses do not say ‘financial firms’ as your sample consists of only banks, while financial firms a term that cover banks, insurance firms, real estate firms and many diversified financial firms.

Again! Your literature review section and the way of developing hypotheses are well structured. However, the literature is outdated in an annoying manner. Please please please update the reviewed literature and you will find new perspective corporate governance.

The paper lacks theorization. This is a major flaw in the study, try to incorporate agency theory.

Please provide a table that highlights how the number of observation were changed after applying each selection criteria for the selected sample. Another table is required that shows the distribution of the 65 banks over the countries and between Islamic and commercial banks.

In table 1 first column, please write the names of the variables correctly and do not use shortcuts. For example, what do you mean by CEOT? Please adjust the names of the variables in the whole manuscript.

The conclusion section is very weak, a conclusion section should show the study implications, contributions, limitations and guidelines for future researchers.

Author Response

Reviewer 3

The paper examines a well-researched factors in the corporate governance filed. However, I am convinced that the large sample size and the comparison between the commercial banks and Islamic banks constitute an original contribution for this research. Therefore, I would suggest the following comments for your consideration.

Response: thanks for your constructive comments. We improved the paper following your guidance. Please find our responses to the comments below.

  • When you mention acronyms, be sure that it has been fully defined earlier in the manuscript (e.g., IBs). Make sure that you follow the journal requirement in presenting the manuscript.

Response: we thank the reviewer for this comment. We make sure to amend this across the paper.

The abstract should capture the research implications, in short words.

Response: Thanks for your comment. Please see the abstract.

Add MENA region to the keywords.

Response: Done

The introduction section is well written, however, you have to use recent literature. The literature you used are so outdated.

Response: Many thanks for this comment. We amend the paper as suggested. Please see section 1 highlighted in red.

In your hypotheses do not say ‘financial firms’ as your sample consists of only banks, while financial firms a term that cover banks, insurance firms, real estate firms and many diversified financial firms.

Response: Done

Again! Your literature review section and the way of developing hypotheses are well structured. However, the literature is outdated in an annoying manner. Please please please update the reviewed literature and you will find new perspective corporate governance.

Response: we thank the reviewer for this comment, we amend the LR section and include some recent literature in both introduction and LR. Please see section 2 highlighted in red.

The paper lacks theorization. This is a major flaw in the study, try to incorporate agency theory.

Response: we added discussion about agency theory as suggested. Please see section 2.1. page 3 highlighted in red.

Another table is required that shows the distribution of the 65 banks over the countries and between Islamic and commercial banks.

Response: Following the reviewer suggestion, we added additional table showing comparisons for the key variables between CBs and IBs using t-test. Please see appendix A1- panel A and Panel B and the discussion of this table in section 3.1 highlighted in red.

In table 1 first column, please write the names of the variables correctly and do not use shortcuts. For example, what do you mean by CEOT? Please adjust the names of the variables in the whole manuscript.

Response: done.

The conclusion section is very weak, a conclusion section should show the study implications, contributions, limitations, and guidelines for future researchers.

Response: We thank the reviewer for this comment. We amend the conclusion as suggested. please see section 5, heighted in red.

Round 2

Reviewer 2 Report

Comments and Suggestions for Authors

some of my comments not achieve. 

some citation no find in the reference list. 

there is no reply to my  previous comments 

Comments on the Quality of English Language

some of my comments not achieve. 

some citation no find in the reference list. 

there is no reply to my  previous comments 

Author Response

Reviewer 2

some of my comments not achieve. 

some citation no find in the reference list. 

there is no reply to my  previous comments 

Response: We thank the reviewer for his comments. We think there is misunderstanding regarding our amendments to the previous reviewer comments since the reviewer mentioned that "there is  no reply to my previous comments". it might be that the reviewer open the old version. please find below our previous  amendments to the reviewer comments .

Reviewer 2

1- The Introduction section is good, but it does not indicate the samples used, the timeframe of the study, or the main findings of the paper. In research papers, the Introduction should summarise the research, including the primary issue or point of contention and the research aims/objectives, contribution, motivation, methods, and findings.

Response: we thank the reviewer for this comment, we amend the introduction to make the motivation and the story of the paper more pronounced. Please see section 1 highlighted in red

2- Literature Review is very important in research:  This section is one of the most critical sections of the research that shows the research gap and helps the researcher cover this gap the researcher failed to list the research gap, shows the importance of research and the means to find solutions to the research problem. I recommend that researchers repeat this section in a professional manner. Moreover, the authors did not consider recent studies related to the topic like 2021,2022, 2023. Therefore, i strongly recommend that the authors rearrange and support this section.

Response: we thank the reviewer for this comment, we amend the LR section and include some recent literature in both introduction and LR. Please see section 2 highlighted in red

3- A discussion of the results is required. Why is an emerging economy and not any other country in the region that has similar characteristics (or is this country an individual compared to others in the region)? In conclusion, the author(s) did not provide consistent arguments to support empirical research on emerging economies in this field, which seriously affects the value of this study.

Response: Following the reviewer suggestion, we added additional table showing the countries characteristics. Please see appendix A1- panel B and the discussion of this table in section 3.1 highlighted in red.

4- The Conclusion section is clear, but it would be better if the authors further expanded the conclusion by offering answers to the research questions on the paper. They then discuss the implications of the study (methodological, managerial, policymakers, and investors). In addition, they must discuss the research limitations in detail and provide an outline for future research.

Response: We thank the reviewer for this comment. We amend the conclusion as suggested. please see section 5, highlited in red

5- In the conclusions, i introduce the recent updated bibliography and comment on it in light of the literature. In this regard, please include the following.

(2022). Do corporate governance and top management team diversity have a financial impact among financial sector? A further analysis. Cogent Business & Management, 9(1), 2141093.

(2023). Board Characteristics and performance of listed firms in Ghana. Corporate Governance: The International Journal of Business in Society23(1), 43-71.

(2022). Board of directors’ attributes and corporate outcomes: A systematic literature review and future research agenda. International Review of Financial Analysis84, 102424.

 Response: we include the suggested references in the paper.

6- Finally, I am not a native speaker, but I suggest author/s to perform professional proofreading.

Comments on the Quality of English Language

Finally, I am not a native speaker, but I suggest author/s to perform professional proofreading.

Response: We proofread the paper again using the MDPI English services. Please see the certificate attached.

Reviewer 3 Report

Comments and Suggestions for Authors

None

Author Response

We appreciate the reviewer efforts in providing constructive comments to increase the paper quality. We thank the reviewer for finding that the paper is reached sufficient level. 

Round 3

Reviewer 2 Report

Comments and Suggestions for Authors

no have 

Author Response

We thanks the reviewer  that he found the paper suitable for publication